



# Sulfuric acid in the Amazon Basin: Measurements and evaluation of existing sulfuric acid proxies

Deanna C. Myers[1], Saewung Kim[2], Steven Sjostedt[3], Alex B. Guenther[2], Roger Seco[4], Oscar Vega Bustillos[5], Julio Tota[6], Rodrigo A. F. Souza[7], and James N. Smith[1]

[1]Department of Chemistry, University of California, Irvine, USA
[2]Department of Earth System Science, University of California, Irvine
[3]Morgan Community College, Fort Morgan, CO, USA
[4]Institute of Environmental Assessment and Water Research (IDAEA-CSIC), Barcelona, Catalonia, Spain
[5]Instituto de Pesquisas Energéticas e Nucleares, Cidade Universitaria, São Paulo, Brazil
[6]Universidade Federal do Oeste do Pará, Santarém, Brazil
[7]Escola Superior de Tecnologia, Universidade do Estado do Amazonas, Manaus, Brazil

**Correspondence:** James N. Smith (jimsmith@uci.edu)

**Abstract.** Sulfuric acid is a key contributor to new particle formation, though measurements of its gaseous concentrations are difficult to make. Several parameterizations to estimate sulfuric acid exist, all of which were constructed using measurements from the Northern Hemisphere. In this work, we report the first measurements of sulfuric acid from the Amazon Basin. These measurements are consistent with concentrations measured in Hyytiälä, Finland, though unlike Hyytiälä there is no clear
correlation of sulfuric acid with global radiation. There was a minimal difference in sulfuric acid observed between the wet and dry seasons in the Amazon Basin. We also test the efficacy of existing proxies to estimate sulfuric acid in this region. Our results suggest that nighttime sulfuric acid production is due to both a stabilized Criegee intermediate pathway, and oxidation of $SO_2$ by OH, the latter of which is not currently accounted for in existing proxies. These results also illustrate the drawbacks of the common substitution of radiation for OH concentrations. None of the tested proxies effectively estimate sulfuric acid
measurements at night. For estimates at all times of day, a recently published proxy based on data from the boreal forest should be used. If only daytime estimates are needed, several recent proxies that do not include the Criegee pathway are sufficient. More investigation of nighttime sulfuric acid production pathways is necessary to close the gap between measurements and estimates with existing proxies.

## 1 Introduction

Numerous studies have shown that sulfuric acid ($H_2SO_4$) contributes significantly to atmospheric particle concentrations. It has been found to be a key component in the formation of new atmospheric aerosol particles (Almeida et al., 2013; Jen et al., 2016; Fiedler et al., 2005; Korhonen et al., 1999; Kuang et al., 2010; Kulmala et al., 2012, 2004; McMurry et al., 2000; Myllys et al., 2019; Weber et al., 1996, 1997; Yao et al., 2018), and a significant contributor to the growth of new particles (Bzdek et al., 2012; Paasonen et al., 2010; Riipinen et al., 2007; Stolzenburg et al., 2005, 2020; Wehner et al., 2005). New particle
formation (NPF) is a major contributor to global cloud condensation nuclei populations (Gordon et al., 2017; Kerminen et al.,



2012; Spracklen et al., 2008, 2010). Given its importance in atmospheric particle formation and growth, accurate measurements of atmospheric $H_2SO_4$ concentrations are necessary for understanding atmospheric chemical and thermal processes and accurately simulating new particle formation (Dunne et al., 2016). However, this has been difficult to achieve because of low ambient concentrations ($10^6$–$10^7$ molecules cm$^{-3}$ or lower), which can only be measured using specialized instrumentation such as chemical ionization mass spectrometers (CIMS) (Dada et al., 2020; Eisele and Bradshaw, 1993; Jokinen et al., 2012; Mikkonen et al., 2011), and because of challenges in deploying and operating these instruments.

Due to these challenges, several studies have developed parameterizations to serve as proxies for $H_2SO_4$ concentrations using its atmospheric sources and sinks (Lu et al., 2019; Weber et al., 1997; Mikkonen et al., 2011; Petäjä et al., 2009). Using measurements of hydroxyl radical (OH) and sulfur dioxide ($SO_2$), Weber et al. (1997) estimated the daytime concentration of $H_2SO_4$ with known rates of photochemical production and loss by condensation onto existing particle surface area (condensation sink, CS) and showed good agreement with measurements of $H_2SO_4$ concentrations made in Hawaii and Colorado, USA. However, like $H_2SO_4$, OH is difficult to measure due to low concentrations and relatively short atmospheric lifetime (Eisele and Bradshaw, 1993). Since OH is formed via ozone ($O_3$) photolysis by ultraviolet radiation and OH concentration has been found to correlate well with UV radiation (Rohrer and Berresheim, 2006), radiation has replaced OH concentrations in current $H_2SO_4$ proxies. This correlation was confirmed by Petäjä et al. (2009), who estimated concentrations of $H_2SO_4$ in Hyytiälä, Finland using proxies with OH measurements, and UV and global radiation as proxies for OH concentration, and found good agreement between estimated and measured $H_2SO_4$ concentrations using both UV and global radiation as OH substitutes. Because global radiation is more frequently measured than UV radiation, Mikkonen et al. (2011) used global radiation to develop proxies based on CIMS measurements of $H_2SO_4$ made in varying environments throughout North America and Europe. They found that the best approximation for all locations depended mainly on radiation strength, with reduced source dependence on the concentration of $SO_2$, and minimal loss contribution from CS. Mikkonen et al. (2011) attributed the reduced dependence on $SO_2$ and CS to these species representing particulate pollution, which would act as both $H_2SO_4$ and OH sinks. Similarly, a proxy developed using measurements of $SO_2$ concentration, UV radiation, and CS from Beijing, China found that CS plays a relatively minor role in determining concentrations of $H_2SO_4$ except when CS is large (Lu et al., 2019). A high correlation between CS and $SO_2$ concentrations was observed, which Lu et al. (2019), like Mikkonen et al. (2011), attributed to both parameters representing atmospheric pollution. Together, the Mikkonen et al. (2011) and Lu et al. (2019) results demonstrate that using only photochemical production and CS as the source and sink, respectively, of H2SO4 is insufficient to accurately estimate its concentration across a wide range of locations.

More recent work has considered additional sources and sinks for atmospheric $H_2SO_4$ to improve these estimates. In addition to formation by OH oxidation of $SO_2$, several proxies described in Dada et al. (2020) consider the formation of $H_2SO_4$ from $O_3$ oxidation of biogenic alkenes via stabilized Criegee intermediates (sCI) (Mauldin et al., 2012). This production pathway is hypothesized to dominate at nighttime, when OH is a less important oxidant (Mauldin et al., 1998). The loss term in these new Dada et al. (2020) proxies include both condensation sink and the clustering of $H_2SO_4$ monomers to form new atmospheric particles. Through testing for a variety of environments, Dada et al. (2020) developed $H_2SO_4$ parameterizations representing sites with conditions similar to those used to develop and verify these proxies. They suggest comparison of any site's $H_2SO_4$,





OH, $SO_2$, $O_3$, and dominant alkene concentrations, as well as global radiation and CS, to those of the sites studied and use the proxy developed for the environment most similar to that of interest. The (Dada et al., 2020) proxies showed good agreement between the measured and estimated concentrations of $H_2SO_4$ for data from sites used in the proxy construction, but thus far the proxies have been tested on one new environment. Further validation of these proxies is needed by testing them on

measurements from a variety of sites.

Though several of the proxies described earlier considered measurements made in varying environments to develop a robust, generalized $H_2SO_4$ proxy (Dada et al., 2020; Mikkonen et al., 2011), only measurements made in the Northern Hemisphere have been used in their construction. Measurements from the Southern Hemisphere need to be considered in order to develop a proxy that accurately estimates $H_2SO_4$ concentrations globally. The Amazon Basin has been the focus of recent field work,

specifically the Observations and Modeling of the Green Ocean Amazon (GoAmazon2014/5) experiment (Martin et al., 2016), in large part because the biological emissions from the forest contribute significantly to climate and atmospheric composition in South America (Artaxo et al., 2013; Pöschl et al., 2010). This region is characterized by a mixture of pristine biogenic conditions with pollution from Manaus and human activity in the area (Nobre et al., 2016). Natural emissions dominate the wet season (December – May), during which time wet deposition of accumulation mode particles (diameter between $0.1 - 2.5$ $\mu$m)

and coarse mode particles (diameter greater than 2.5 $\mu$m) reduces concentrations of particles in these size ranges compared to the dry season (August - November). However, recent work has shown that anthropogenic pollutants influence atmospheric particles during the wet season as well (Glicker et al., 2019). Previous measurements in the Amazon Basin have reported concentrations of $SO_2$ that were more than an order of magnitude smaller than those measured in remote sites in the Northern Hemisphere (Andreae and Andreae, 1988; Andreae et al., 1990; Martin et al., 2010). From these measurements, model results

have suggested that $H_2SO_4$ levels are too low to result in surface-level particle formation (Spracklen et al., 2006). However, measurements of $H_2SO_4$ levels in the Amazon Basin have not yet been reported.

This manuscript reports the first measurements of $H_2SO_4$ in the Amazon Basin, performed using chemical ionization mass spectrometry. The focus of this work is during two intensive operating periods (IOPs) during the GoAmazon2014/5 campaign; one during the wet season (IOP 1: 9 February 2014 – 8 March 2014) and one during the dry season (IOP 2: 28 August 2014 –

5 September 2014). We then assess the efficacy of existing proxy parameterization in estimating $H_2SO_4$ concentrations in the Amazon Basin, the first location in the Southern Hemisphere to be tested.

## 2 Methods

### 2.1 Site description

All chemical and meteorological measurements were performed during the GoAmazon2014/5 campaign at the T3 site (3.2133°

S, 60.5987° W), 10 km northeast of Manacapuru, Brazil (Martin et al., 2016). This site is located in pastureland 70 km west of Manaus, Brazil, in central Amazonia. Measurement facilities deployed to T3 included the Atmospheric Radiation Measurement (ARM) Mobile Facility number 1 (AMF-1), the ARM Mobile Aerosol Observing System (MAOS), and laboratories contained in four modified shipping containers with instruments operated by several research organizations. Air masses arriving at this





site typically originate near the coast of the Atlantic Ocean and contain biogenic species from the forest as they travel to the
site, with some influence from Manaus. All times are reported in UTC.

## 2.2 Instrumentation

### 2.2.1 Trace Gas Analysis

Gas-phase concentration measurements of $H_2SO_4$ and OH were made using a selected ion chemical ionization mass spectrometer (SICIMS), the details of which have been reported previously in Jeong et al. (2022), Tanner et al. (1997), and Mauldin
et al. (1998). Concentrations of $SO_2$ were measured using a Thermo Fisher Scientific Model 43i trace level-enhanced pulsed fluorescence $SO_2$ analyzer with a detection limit of $\tilde{2}.4 \times 10^8$ molec $cm^{-3}$. More specific information regarding the operation and calibration of the $SO_2$ analyzer can be found in Springston (2016). A Thermo Fisher Scientific Ozone Analyzer Model 49i was used to measure concentrations of $O_3$ based on their absorption of ultraviolet (254 nm) light. More details regarding the operation of this instrument can be found in Springston (2020). Measurements of monoterpene (MT) and isoprene
concentrations were obtained using a selected reagent ion proton transfer reaction time-of-flight mass spectrometer (SRI-PTR-TOFMS). These data were calibrated using the ion signal of $C_{10}H_{17}^+$ for $\alpha$-pinene and $C_5H_9^+$ for isoprene, and $\alpha$-pinene and isoprene standards. More specific details about the operation of this instrument are reported in Sarkar et al. (2020). All trace gas concentrations are reported as five-minute averages with units of molecules $cm^{-3}$.

### 2.2.2 Particle Number-Size Distribution

Particle number-size distributions for particles with electrical mobility diameters 10 – 496 nm from 0:00 UTC 5 February – 18:46 16 February, and 11 – 460 nm for the rest of IOP 1 and IOP 2 were collected using a TSI Model 3963 scanning mobility particle sizer with a TSI Model 3772 condensation particle counter (CPC) (ARM, 2014c). CS was estimated from the number size distributions for particles with mobility diameters 11 – 460 nm using the method described in Kulmala et al. (2001) and Kulmala et al. (2012).

### 2.2.3 Meteorology

Global radiation was measured at the AMF-1 using a precision spectral pyranometer (Eppley) (ARM, 2014b). Data were collected in 60-second intervals. Ambient temperature, relative humidity, wind direction, and wind speed were measured at AMF-1 in 60-second intervals (ARM, 2014a). All meteorological data are reported as 5-minute averages.

### 2.3 Proxies Tested

We used measurements of $SO_2$ and OH along with estimates of CS to evaluate the efficacy of the simplest $H_2SO_4$ proxy developed, which includes the photochemical production of $H_2SO_4$ and loss to particle surface area in estimating the concentration





of $H_2SO_4$ using the following equation:

$$\frac{d[\mathrm{H_2SO_4}]}{dt} = k[\mathrm{OH}][\mathrm{SO_2}] - [\mathrm{H_2SO_4}]CS^{-1} \tag{1}$$

in which $k$ is the temperature-dependent rate constant (DeMore et al., 1997; Sander et al., 2003). Assuming that $H_2SO_4$
production and loss are in steady-state, Eq. (1) can be rearranged to directly calculate the concentration of $H_2SO_4$ (Proxy 1,
Table 1). To evaluate whether global radiation (GlobRad) is an effective replacement for OH concentrations in the Amazon
Basin, we used Proxy 2, where $k'$ replaces the temperature-dependent rate constant $k$, and is the fitting parameter between
the proxy terms and measured concentration of $H_2SO_4$, similar to the proxy reported by Petäjä et al. (2009) (Table 1). We
also used several of the proxies developed from data sets collected at a variety of locations to assess how well they estimate
$H_2SO_4$ concentrations in the Amazon Basin. This includes the proxy Mikkonen et al. (2011) reported that best predicted
$H_2SO_4$ concentrations across all of the locations they tested, where $k$ is the temperature-dependent rate constant for the
reaction of OH with $SO_2$ (DeMore et al., 1997) multiplied by $10^{12}$ (Proxy 3, Table 1). Recent proxies developed by Dada et al.
(2020) that additionally consider $H_2SO_4$ production via the sCI pathway and loss due to clustering were tested to evaluate
the relative importance of these pathways in determining $H_2SO_4$ concentrations in the Amazon Basin. Based on the values
of the characteristic predictor variables ([$H_2SO_4$], [$SO_2$], CS, Global radiation, [$O_3$], [Alkene]) detailed in Figure 9 of that
work, we tested proxies representing environments similar to the boreal forest (Hyytiälä, Finland) (Proxy 4), and representing
environments similar to the rural location (Agia Marina, Cyprus) used to develop this proxy (Proxy 5). Notably, Proxy 4 is the
only proxy tested that includes the sCI production pathway, making it possible to assess nighttime $H_2SO_4$ estimations, one of
the limitations of the proxies that only consider photochemical $H_2SO_4$ production. The equations corresponding to each proxy
(numbered 1 - 5) are shown below in Table 1.

**Table 1.** Proxies used in this study to estimate sulfuric acid concentrations. Parameter terms defined in Section 2.3.

| Proxy | Equation |
|---|---|
| 1 | $[H_2SO_4] = \dfrac{k[OH][SO_2]}{CS}$ |
| 2 | $[H_2SO_4] = \dfrac{k' \cdot GlobRad[SO_2]}{CS}$ |
| 3 | $[H_2SO_4] = 8.21 \times 10^{-3} \cdot k \cdot GlobRad[SO_2]^{-0.62} \cdot (CS \cdot RH)^{-0.13}$ |
| 4 | $[H_2SO_4] = \dfrac{CS}{2 \cdot (4.2 \times 10^{-9})} + \left[ \left( \dfrac{CS}{2 \cdot (4.2 \times 10^{-9})} \right)^2 + \dfrac{SO_2}{4.2 \times 10^{-9}} \left( 8.6 \times 10^{-9} \cdot GlobRad + 6.1 \times 10^{-29}[O_3][Alkene] \right) \right]^{\frac{1}{2}}$ |
| 5 | $[H_2SO_4] = \dfrac{CS}{2 \cdot (2.0 \times 10^{-9})} + \left[ \left( \dfrac{CS}{2 \cdot (2.0 \times 10^{-9})} \right)^2 + \dfrac{SO_2}{2.0 \times 10^{-9}} \left( 9.0 \times 10^{-9} \cdot GlobRad \right) \right]^{\frac{1}{2}}$ |

## 3   Results and Discussion

Table 2 lists the key variables for the proxies used in this study across both IOPs. Due to instrument malfunctions as well as
challenges associated with operating this instrument in this remote location, only a select number of days from each IOP are



**Table 2.** Summary of the mean, median, 5 - 95 percentiles, and standard deviation (sd) of the relevant trace gases, condensation sink, global radiation, and relative humidity measured in the Amazon Basin during this study.

| | | IOP 1 (wet season) | IOP 2 (dry season) | Campaign (combined) |
|---|---|---|---|---|
| $[H_2SO_4]\ 10^5$ | Mean | 9.53 | 3.85 | 7.89 |
| | Median | 7.82 | 2.59 | 6.73 |
| molec cm⁻³ | 5 − 95 percentiles | 5.17 − 20.4 | 1.05 − 10.8 | 1.66 − 18.7 |
| | Sd | 5.01 | 3.19 | 5.21 |
| $[OH]\ 10^5$ | Mean | 11.1 | 3.85 | 7.78 |
| | Median | 9.49 | 2.64 | 6.86 |
| molec cm⁻³ | 5 − 95 percentiles | 5.42 − 21.9 | 0.41 − 11.0 | 0.63 − 20.2 |
| | Sd | 5.23 | 3.49 | 5.79 |
| $[SO_2]\ 10^9$ | Mean | 1.41 | 2.29 | 1.95 |
| | Median | 1.23 | 1.98 | 1.58 |
| molec cm⁻³ | 5 − 95 percentiles | 0.51 − 3.55 | 0.94 − 6.08 | 0.62 − 5.02 |
| | Sd | 1.16 | 1.83 | 1.72 |
| CS $10^{-3}$ | Mean | 5.45 | 18.7 | 11.6 |
| | Median | 4.81 | 17.0 | 7.54 |
| s⁻¹ | 5 − 95 percentiles | 1.21 − 11.8 | 5.13 − 38.7 | 1.56 − 31.7 |
| | Sd | 4.77 | 12.1 | 11.1 |
| Radiation ( > 10) | Mean | 614 | 666 | 636 |
| | Median | 512 | 646 | 587 |
| W m⁻² | 5 − 95 percentiles | 39 − 1460 | 57 − 1270 | 43 − 1370 |
| | Sd | 465 | 393 | 437 |
| $[O_3]\ 10^{11}$ | Mean | 2.22 | 4.43 | 3.14 |
| | Median | 1.86 | 3.66 | 2.21 |
| molec cm⁻³ | 5 − 95 percentiles | 0.40 − 5.09 | 0.36 − 11.2 | 0.38 − 9.25 |
| | Sd | 1.74 | 3.62 | 2.89 |
| $[Isoprene]\ 10^{10}$ | Mean | 1.82 | 3.15 | 1.98 |
| | Median | 1.10 | 1.98 | 1.62 |
| molec cm⁻³ | 5 − 95 percentiles | 0.68 − 6.32 | 0.72 − 10.2 | 0.70 − 7.18 |
| | Sd | 2.16 | 4.01 | 2.77 |
| RH (%) | Mean | 90.5 | 82.5 | 88.5 |
| | Median | 95.3 | 88.6 | 94.2 |
| | 5 − 95 percentiles | 66.5 − 99.6 | 52.9 − 99.5 | 99.6 |
| | Sd | 10.7 | 16.1 | 12.8 |

included for analysis. The measurements reported here span 14 days across IOP 1 (9 - 19 February and 5 - 8 March 2014) and 9 days across IOP 2 (28 August - 5 September); thus the campaign data is more representative of measurements made during IOP 1 (61 % of the total data points). Measurements of $H_2SO_4$ during both IOPs show a small degree of seasonality (IOP 1 median: 7.82 x $10^5$ molec cm⁻³; IOP 2 median: 2.56 x $10^5$ molec cm⁻³), indicating that differences between the wet (IOP 1) and dry (IOP 2) seasons do not influence $H_2SO_4$ to a large degree. The campaign median value (6.73 x $10^5$ molec cm⁻³) is consistent with measurements from Hyytiälä, Finland (Dada et al., 2020; Mikkonen et al., 2011), which suggests that the boreal forest environment may be similar enough to use the boreal proxy reported in Dada et al. (2020). Measured $H_2SO_4$ is roughly one order of magnitude lower than measurements from rural (Agia Marina, Cyprus (Dada et al., 2020)), semi-urban (Helsinki, Finland (Dada et al., 2020)), urban (Atlanta, USA (Mikkonen et al., 2011); Budapest, Hungary (Dada et al., 2020)), and megacity (Beijing, China (Dada et al., 2020; Lu et al., 2019; Yang et al., 2021)) environments. These data suggest that the





general proxy from Mikkonen et al. (2011) (Proxy 3) and boreal Dada et al. (2020) proxy (Proxy 5) may provide reasonable

estimations.

Measurements of $SO_2$ and $O_3$ similarly show minimal differences between the wet and dry seasons. Observed $SO_2$ concentrations are consistent with measurements from boreal (Hyytiälä, Finland (Dada et al., 2020; Mikkonen et al., 2011)), alpine forest (Niwot Ridge, USA (Mikkonen et al., 2011)), and rural (Agia Marina, Cyprus (Dada et al., 2020); San Pietro Capofiume, Italy (Mikkonen et al., 2011)) environments. The observed levels of $SO_2$ are consistent with the smaller industrial influence in

the Amazon Basin than in semi-urban (Helsinki, Finland (Dada et al., 2020)), urban (Budapest, Hungary (Dada et al., 2020); Atlanta, USA (Mikkonen et al., 2011)), and megacity (Beijing, China (Dada et al., 2020)) locations. Measurements of $O_3$ concentrations during both IOPs are lower than those reported for all sites used in the Dada et al. (2020) and Mikkonen et al. (2011) studies, although measurements from Hyytiälä reported in both studies are the closest to our measurements.

Unlike $H_2SO_4$, $SO_2$, and $O_3$, measurements of CS are consistent with previous observations of more polluted conditions

during the dry season (IOP 2) compared to the wet season (IOP 1) (Andreae et al., 2004; Rcia et al., 2000), though measurements of $SO_2$ concentration suggest this difference is not to the degree previously hypothesized (median 0.73 x $10^9$ for IOP 1, 1.48 x $10^9$ molec cm$^{-3}$ for IOP 2). During IOP 1, the median measured CS (4.81 x $10^{-3}$ s$^{-1}$) is consistent with previously reported values from several types of forest (Hyytiälä, Finland (Dada et al., 2020; Mikkonen et al., 2011); Niwot Ridge, USA (Mikkonen et al., 2011)) and rural (Agia Marina, Cyprus; San Pietro Capofiume, Italy (Dada et al., 2020)) environments. The

larger CS measured during IOP 2 (median: 17.0 x $10^{-3}$ s$^{-1}$) is more consistent with those reported from Atlanta (Mikkonen et al., 2011), and even the megacity Beijing (Dada et al., 2020). This difference in CS between the two seasons is mainly driven by a the higher concentration of 50 - 100 nm particles present during IOP 2 (average: 1530 cm$^{-3}$) and not IOP 1 (average: 300 cm$^{-3}$) (Fig. S1), and is consistent with the increased wet deposition of particles during the wet season (IOP 1) (Andreae et al., 2004; Rcia et al., 2000). The campaign-averaged CS (7.54 x $10^{-3}$ s$^{-1}$) is consistent with prior measurements from San Pietro

Capiofume, Italy (Mikkonen et al., 2011) and Budapest, Hungary (Dada et al., 2020). The CS measurements support the use of the (Mikkonen et al., 2011) proxy and the (Dada et al., 2020) boreal and rural proxies.

We compared the concentrations of isoprene and monoterpenes to determine the dominant alkene, which was used in the Dada et al. (2020) boreal proxy (Proxy 4), per the recommendation in that study. Isoprene was observed to have a higher concentration (campaign median: 1.62 x $10^{10}$ molec cm$^{-3}$) than monoterpenes (campaign median: 3.33 x $10^9$ molec cm$^{-3}$),

and was thus used in the Dada et al. (2020) boreal estimation as the alkene concentration. The isoprene concentrations measured during the campaign were about an order of magnitude greater than measured monoterpene levels from Hyytiälä, and significantly lower than alkene concentrations measured in Beijing (Dada et al., 2020), supporting the use of the Dada et al. (2020) boreal proxy. The levels of these key variables (CS, $H_2SO_4$, $SO_2$, $O_3$, and isoprene) in estimating the concentration of $H_2SO_4$ in the Amazon Basin show that the generalized Mikkonen et al. (2011) proxy and both the boreal and rural Dada et al.

(2020) proxies may be appropriate to use in this location.

Next, we compared the two-hour diurnal cycles of the source terms ($SO_2$, OH, and radiation) in the basic photochemical proxies to assess their correlation with the measured concentrations of $H_2SO_4$ (Fig. 1). There is no apparent diurnal cycle of $H_2SO_4$ and, notably, there is not a clear correlation between its concentration and the level of global radiation measured at the





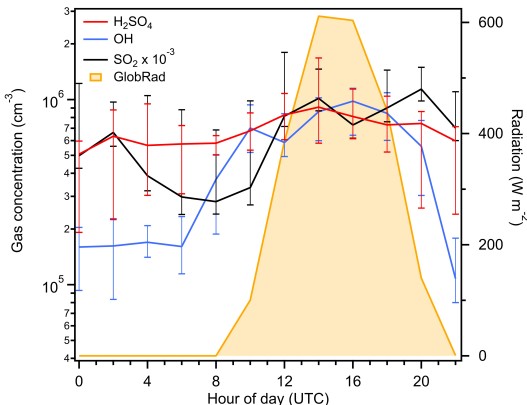

**Figure 1.** Two-hour diurnal variation of the median $H_2SO_4$, $SO_2$, OH, and global radiation measured during the entire campaign. Note that daylight hours are from 08:00 - 22:00 UTC during the campaign; negligible changes between IOPs 1 and 2 were observed.

site. This is in contrast to the correlation observed between these two parameters at the Northern Hemisphere sites used in the
construction of the Mikkonen et al. (2011) and Dada et al. (2020) proxies (data sets from Atlanta, USA; Hyytiälä, Finland; Melpitz, Germany; Niwot Ridge, USA). During the observation period, 36 % of the total $H_2SO_4$ was measured at night, suggesting that while photochemical production is likely an important source of $H_2SO_4$ Amazon Basin, nighttime sources should also be considered in an efficient proxy.

Additionally, Figure 1 shows that there was OH measured during nighttime (22:00 - 08:00 UTC). This suggests that the
common use of global radiation as an OH replacement in $H_2SO_4$ proxies is only sufficient during daytime hours (08:00 - 22:00 UTC) in the Amazon Basin. This is consistent with model results from Lelieveld et al. (2008, 2016), which indicate that secondary production of OH through $O_3$ reaction with isoprene is a major source of OH in the boundary layer in the Amazon rainforest, in addition to primary production from photodissociation of $O_3$. This secondary pathway is active at nighttime, and likely contributes in other regions where data sets have been used to construct and test $H_2SO_4$ proxies, meaning that nighttime
$H_2SO_4$ is not being accounted for in these estimations. Thus, as we move through our testing of the proxies that substitute global radiation for OH, it is with the understanding that this substitution misses nighttime production of $H_2SO_4$ through the oxidation of $SO_2$ by OH, which is likely occurring in this location.

The concentration of $H_2SO_4$ was estimated using Proxy 1, which includes production from the oxidation of $SO_2$ by OH and loss from CS. The results of this estimation are plotted as a function of the measured $H_2SO_4$ in Figure 2a. Estimates
from IOPs 1 and 2 fall below the 1:1 line, meaning the proxy tends to underestimate measured $H_2SO_4$ by an average factor of 3.7. Despite a generally linear trend exhibited between the estimated and measured values, there is a weak correlation (0.46) between these two that cannot be attributed to a single parameter (CS, OH, $SO_2$) included in the proxy. While this proxy is advantageous in that it is the only proxy tested that depends directly on the concentrations of species that react to form $H_2SO_4$ and uses measured rate constants to perform estimations, in the Amazon Basin this estimation provides a lower limit
of $H_2SO_4$ concentrations. Our results further support the hypothesis that there is another source of $H_2SO_4$ in this region that





is not described by OH initiated oxidation of $SO_2$. They also indicate that loss from CS may not be the only loss pathway for $H_2SO_4$.

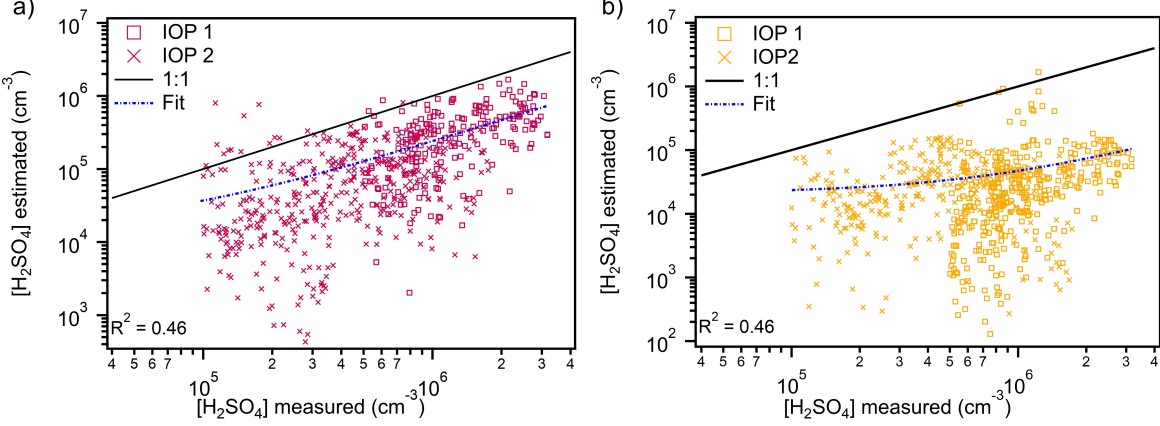

**Figure 2.** Estimated concentrations of sulfuric acid from Proxy 1 (865 points) (a) and Proxy 2 (1941 points) (b) versus measured concentrations. Data from IOP 1 are plotted as boxes and data from IOP 2 are plotted as crosses. The 1:1 line is plotted to guide the eye. The fit line represents the fit between the measured and proxy-estimated values of sulfuric acid.

To evaluate whether global radiation is a sufficient substitute for OH during daytime, we used Proxy 2 to estimate $H_2SO_4$. The value of $k'$ was calculated as a fit parameter between the log of the proxy terms (GlobRad, $SO_2$, CS) and the log of the measured $H_2SO_4$ for the entire data set (Fig. S2). The calculated value of $k'$ is 2.43 x $10^{-10}$ m$^2$ s$^1$ W$^{-1}$, which is smaller than the fit value reported in Petäjä et al. (2009) (1.4 x $10^{-7}$ m$^2$ s$^1$ W$^{-1}$). The difference in $k'$ is a result of the dependence of the proxy on radiation between the location used in this study and Hyytiälä, which was used in Petäjä et al. (2009). A drawback to this estimation compared to Proxy 1 is that it does not rely on the specific reactants that produce $H_2SO_4$. Figure 2b shows that this estimation, like that from Proxy 1, falls below the 1:1 line, though to an even larger degree than the first proxy. Measurements of OH and radiation show little correlation during the observation period (Fig. S3), supporting the hypothesis that secondary OH production from $O_3$ reaction with isoprene contributes significantly in this region (Lelieveld et al., 2008, 2016). Similar results are obtained when using the proxy reported by Petäjä et al. (2009) (Fig. S4). Both proxies do a particularly poor job estimating concentrations during IOP 2 (Fig. 2b), in which the estimates do not exhibit a trend with the measured values. This can be attributed to a lack of correlation between $H_2SO_4$ and radiation during this portion of the observation period (Fig. S3a).

Interestingly, the main underestimations made with Proxy 2 occur when the value of global radiation falls between 10 - 100 W m$^{-2}$. Previous studies have used 10 W m$^{-2}$ (Mikkonen et al., 2011) and 50 W m$^{-2}$ (Dada et al., 2020) as the lower cut-off for radiation, although these results indicate that increasing the lower limit for radiation to 100 W m$^{-2}$ would likely improve estimates. Since both $H_2SO_4$ and OH were measured when radiation was less than 100 W m$^2$ throughout the entire campaign (Fig. S3), this would be at the expense of estimating $H_2SO_4$ during low-light (radiation < 100 W m$^{-2}$) conditions,



when secondary production of OH is likely the dominant source of OH. This discrepancy suggests that a combination of other $H_2SO_4$ sources and secondary OH production are contributing to $H_2SO_4$ levels, which is not being accounted for in this parameterization. Further investigation into the relative importance of primary and secondary OH production pathways should be performed to determine a generalized radiation lower cut-off value for application of these general $H_2SO_4$ proxies during daytime hours. Additionally, more examination of the relative contributions from primary and secondary OH production pathways is necessary to evaluate how well solar radiation represents OH across a range of locations.

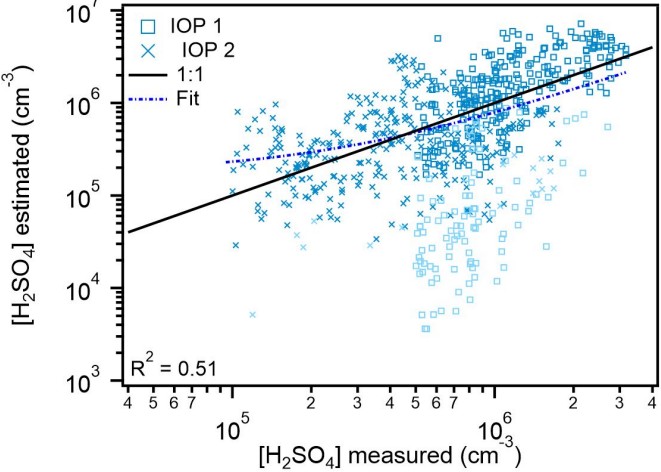

**Figure 3.** Estimated concentrations of sulfuric acid from Proxy 3 versus measured concentrations (1172 points). Data from IOP 1 is plotted as boxes and data from IOP 2 is plotted as crosses. Data points are color-coded to represent the amount of global radiation measured at that time; light blue points were when global radiation was 0 - 100 W m$^2$, and dark blue points were when global radiation exceeded 100 W m$^2$. The 1:1 line is plotted to guide the eye. The fit line represents the fit between the measured and proxy-estimated values of sulfuric acid.

The best predictive proxy reported in Mikkonen et al. (2011) (Proxy 3) was also tested using the Amazon Basin data set. Like Proxy 2, this uses global radiation instead of OH, though as described earlier it was developed using measurements from a variety of different environments and has significant differences in both the $H_2SO_4$ source and sink terms. This proxy has a reduced dependence on $SO_2$ in the source term, as well as a reduced dependence on loss to particle surface area, which includes a term meant to represent particulate hygroscopic growth ($CS \cdot RH$) (Table 1). Figure 3 shows that the estimations from both IOPs fall much closer to the 1:1 line than for Proxies 1 and 2, with a particularly noticeable improvement for IOP 2 compared to Proxy 2. Unlike with Proxy 2, the estimations here for IOP 2 exhibit a trend with the measured values of $H_2SO_4$. The lighter-colored markers represent data points where global radiation is between 10 - 100 W m$^{-2}$. This underestimation during these low-light conditions was also seen in the estimates from Proxy 2, further supporting the need for inclusion of secondary OH production in an effective parameterization in the Amazon Basin and more investigation into a generalized lower limit for values of radiation used in these parameterizations. These improved estimates from this proxy with reduced dependence on the concentration of $SO_2$ support the hypothesis reported in Mikkonen et al. (2011) that $SO_2$ is an indicator of particulate





pollution, which acts as a sink for both $H_2SO_4$ and OH. Additionally, the Amazon Basin is very humid (campaign average RH

$89 \pm 13$ %), so accounting for hygroscopic growth of particles in the CS term may better represent the actual particle surface area available for $H_2SO_4$ uptake. This can also help explain the marked improvements over estimates from Proxies 1 and 2, both of which underestimate measured $H_2SO_4$.

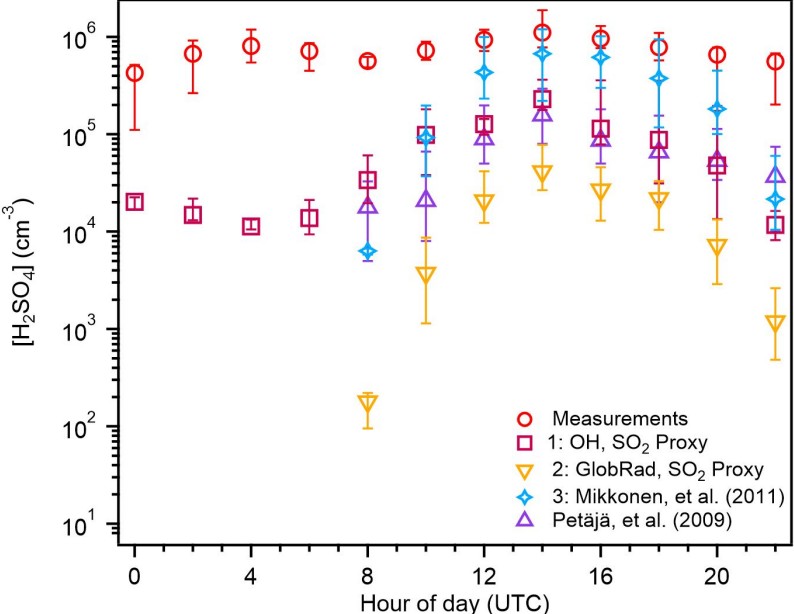

**Figure 4.** Two-hour averaged diurnal variation of the median sulfuric acid measurements (red), and estimations from Proxies 1 (purple), 2 (yellow), and 3 (blue) for the entire campaign. The bars represent the $25^{th}$ - $75^{th}$ percentiles for each measured value. Daylight hours: 08:00 - 22:00 UTC.

    We plotted the diurnal cycle of each proxy to assess their efficacy in estimating $H_2SO_4$ at different times of the day (Fig. 4). Proxy 1, which is the only proxy to include the concentration of OH, is also the only proxy shown to include nighttime

estimations of $H_2SO_4$. Since both species were measured at night in the Amazon Basin (Fig. 1), this illustrates a major limitation of the other proxies that use global radiation as a substitute for OH. Despite Proxy 1 providing nighttime estimates of $H_2SO_4$, it tends to under-predict measurements by an order of magnitude during these hours. When radiation exceeds 100 W m$^{-2}$ (10:00 UTC, Fig. 1), the proxy reported by Petäjä et al. (2009), which is very similar to Proxy 2 in this work, is competitive with Proxy 1 in its predictive ability, while Proxy 2 is within the $25^{th}$ percentile of the Petäjä et al. (2009) estimation, and

Proxy 3 underestimates the measured values by two orders of magnitude. From 12:00 - 20:00 UTC, the Mikkonen et al. (2011) proxy (Proxy 3) best estimates the measured concentrations of $H_2SO_4$; the median estimation falls within the $25^{th}$ - $75^{th}$ percentiles of the measured values. Proxy 1 and the Petäjä et al. (2009) proxy underestimate measured concentrations by one order of magnitude during this time period, while Proxy 2 underestimates by $10^1$ - $10^2$ molec cm$^{-3}$. During daylight hours, Proxies 1 and 3 are sufficient estimators of $H_2SO_4$ while Proxy 2 drastically underestimates measurements. Only Proxy 1 can




provide nighttime estimations, which are necessary in the Amazon Basin where $H_2SO_4$ is measured at night. This proxy is the only one tested thus far that accounts for secondary OH production.

Several new proxies reported by Dada et al. (2020) include production of $H_2SO_4$ through a sCI pathway, as well as an additional loss pathway due to clustering of $H_2SO_4$ to form new particles. This additional source of $H_2SO_4$ is active at nighttime, so despite these proxies depending on global radiation rather than measurements of OH concentration (Proxies 4 and

5, Table 1), nighttime estimations can still be made. Based on Figure 9 of Dada et al. (2020), the proxies developed representing boreal forest and rural environments would be most appropriate to use for the Amazon Basin conditions. Of the two proxies, only the boreal (Proxy 4) includes the sCI production pathway, though both proxies include the clustering loss term. The rural proxy (Proxy 5) can therefore be compared to Proxies 1 - 3 to evaluate the best predictive daytime parameterization for the Amazon Basin.

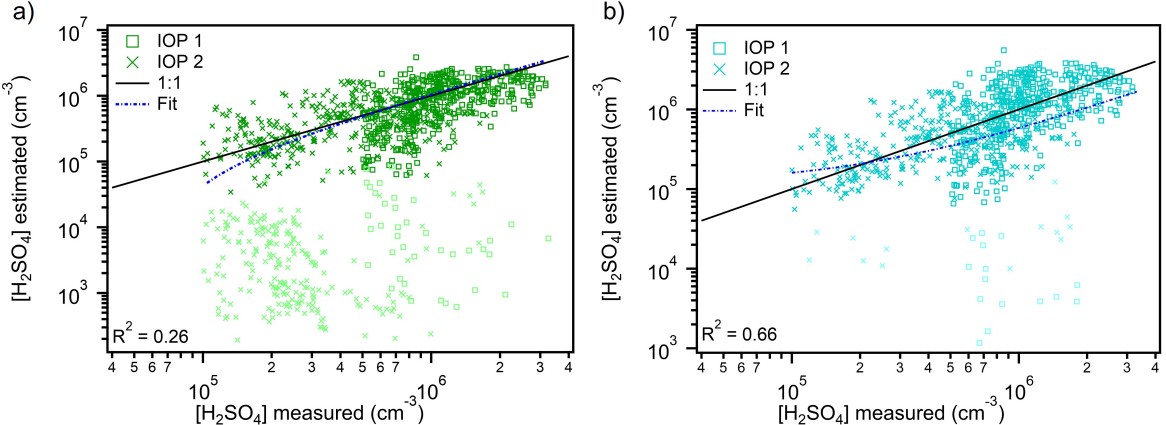

**Figure 5.** Estimated concentrations of sulfuric acid from Proxy 4 (1941 points) (a) and Proxy 5 (1654 points) (b) versus measured concentrations. Data from IOP 1 is plotted as boxes and data from IOP 2 is plotted as crosses. Data points are color-coded to represent the amount of global radiation measured at that time; lighter-colored points were when global radiation was 0 - 100 W $m^2$, and darker-colored points were when global radiation exceeded 100 W $m^2$. The 1:1 line is plotted to guide the eye. The fit line represents the fit between the measured and proxy-estimated values of sulfuric acid.

Figure 5a shows that data points where global radiation exceed 100 W $m^{-2}$ from the boreal proxy (Proxy 4) fall on the 1:1 line, while those from lower-light conditions all underestimate the measured values. These underestimations ($10^1$ - $10^2$ molec $cm^{-3}$) represent data points from both nighttime and twilight times of day, and are likely due to the proxy only considering the sCI formation pathway during these times. The weak correlation (0.26) between the estimated and measured values is driven by the low-light data points; a much higher correlation (0.68) is achieved for data points where global radiation > 100

W $m^{-2}$. Since OH was measured during nighttime in the Amazon Basin, the production of $H_2SO_4$ from OH oxidation of $SO_2$ is an unaccounted for source in this estimation, and likely contributes to the low-light underestimations observed. Similar results were obtained using the combined concentrations of isoprene and monoterpene as the alkene term in this proxy (Figs.





S5 and S6). Interestingly, the nighttime $H_2SO_4$ production term in this proxy likely also represents the main secondary OH production pathway (Table 1). This illustrates the need to distinguish boreal forest environments from the tropical rainforest due to differences in OH sources; model results suggest that primary production of OH and secondary production due to $NO_x$ are more important in the boreal forest than tropical rainforest (Lelieveld et al., 2016). The Lelieveld et al. (2016) results also indicate that even during summertime, nighttime OH is lower in the boreal forest than in the tropical rainforest. As pollution, including NOx, is expected to increase in the Amazon Basin, model results made from GoAmazon2014/5 data suggest that OH levels will increase (Liu et al., 2018). Despite the similarity in many of the $H_2SO_4$ key predictor variables between the Amazon Basin and Hyytiälä, there are major differences between these two locations that require consideration when using Proxy 4.

Proxy 5, which is representative of rural conditions, does not include the sCI pathway and therefore only provides daytime estimations of $H_2SO_4$. Data from both IOPs lie near the 1:1 line, though they have more spread around this line than the daytime estimations from Proxy 4 (Fig. 5b). The few low-light data points used in this parameterization exhibit the underestimation trend seen in Proxies 3 and 4, likely due to a combination of missing the sCI $H_2SO_4$ source and secondary OH production like Proxy 3. There is a clear improvement in the predictive strength of this estimation compared to Proxy 1, which almost entirely underestimates measured concentrations of $H_2SO_4$ (Fig. 2a).

Both of the Dada et al. (2020) proxies have a higher correlation with measured $H_2SO_4$ when global radiation exceeds 100 W m$^{-2}$ (Fig. 5) than the other radiation-based proxies (Fig. 3 and 4). This suggests that Proxies 4 and 5 should have daytime estimations that are more consistent with the Amazon Basin measurements than the previous proxies. Additionally, Proxy 4 should provide estimates during all hours of the day. To test this hypothesis, the diurnal cycles of these proxies and the measurements of $H_2SO_4$ were plotted for comparison.

As hypothesized, the estimations from 12:00 - 20:00 UTC for Proxy 4 and 14:00 - 20:00 UTC for Proxy 5 are within the $25^{th}$ - $75^{th}$ percentile bars of the $H_2SO_4$ measurements (Fig. 6). Both estimations at 10:00 UTC are similar to those from Proxies 1 and 3, and all four estimate more accurately than Proxy 2 and the Petäjä et al. (2009) proxy (Fig. 4). The consistency between Proxies 3 and 5 during daylight hours indicates that the clustering of $H_2SO_4$ molecules to form new atmospheric particles is not a major loss source during this time of day. The boreal proxy (Proxy 4) greatly underestimates measurements at nighttime ($10^2$ molec cm$^{-3}$), and are one order of magnitude smaller than those from Proxy 1 (Fig. 4). In order to match the concentrations of $H_2SO_4$ measured between 0:00 - 8:00 UTC, there would need to be an increase of $10^3$ molec cm$^{-3}$ of alkene (median concentration necessary: 2.9 x $10^{12}$ molec cm$^3$), which is larger than the total concentration of monoterpenes and isoprene measured during the campaign (Fig. S6). These results suggest that both the sCI and OH oxidation of $SO_2$ may be contributors at nighttime in the Amazon Basin, and perhaps in other locations as well. Estimating $H_2SO_4$ concentrations at night is currently the main area of uncertainty with current proxies, and while measurements of OH are difficult to make, they are key to determining low-light and nighttime sources of $H_2SO_4$ for developing a robust proxy for general use.

The boreal proxy from Dada et al. (2020) (Proxy 4) is the best general use proxy for the Amazon Basin. This proxy provides the most representative estimations of $H_2SO_4$, considering both overall estimations (Fig. 5) and the diurnal cycle compared to measured values (Fig. 6). Though both the Mikkonen et al. (2011) and rural proxies provide similarly accurate estimations





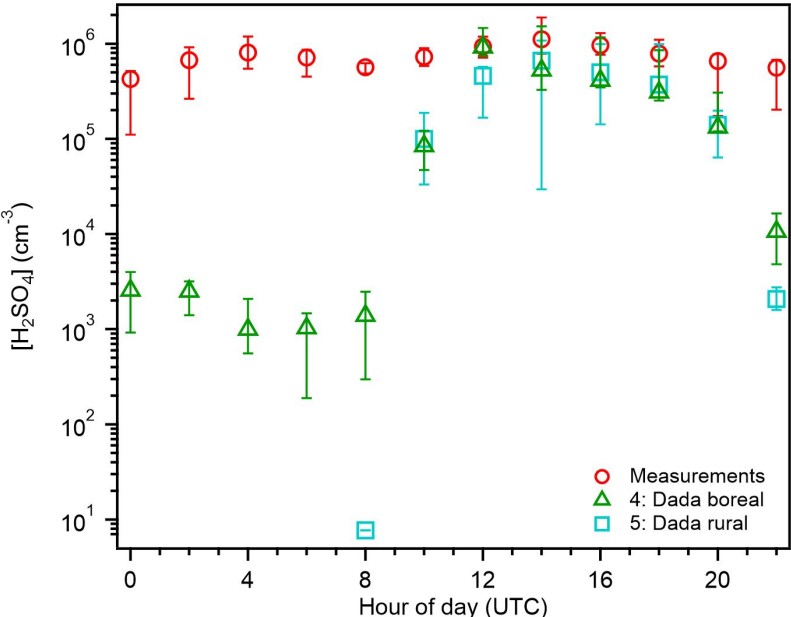

**Figure 6.** Two-hour averaged diurnal variation of the median sulfuric acid measurements (red), and estimations from Proxies 4 (green), and 5 (teal) for the entire campaign. The bars represent the $25^{th}$ - $75^{th}$ percentiles for each value. Daylight hours: 08:00 - 22:00 UTC.

during daylight hours, Proxy 4 is the only one of these three to include nighttime estimations. Our results support the Dada et al. (2020) recommendation to compare a given location's conditions to those reported in Figure 9 of that work to determine

315 the most appropriate proxy to use. The conditions in the Amazon Basin best aligned with the boreal conditions reported in that work, and that proxy provided the best estimates of $H_2SO_4$. We note that caution should be applied to estimates from this parameterization due to differences in OH production pathways between the boreal forest and tropical rainforest environments. These results support the inclusion of the sCI production pathway and loss due to clustering pathway in a robust proxy. They also show that replacing the concentration of OH with global radiation is insufficient for proxies in the Amazon Basin where

320 OH has been measured at nighttime (Fig. 1), and likely contributes to the measured $H_2SO_4$ during this time of day. For estimations of solely daytime concentrations of $H_2SO_4$ (global radiation > 100 W m$^{-2}$), the Dada et al. (2020) estimations (Proxies 4 and 5) and Mikkonen et al. (2011) parameterization (Proxy 3) provide the best estimations of $H_2SO_4$ (Figs. 3-6). These proxies provide better daytime estimations than the photochemical proxies that only consider production of $H_2SO_4$ via OH oxidation of $SO_2$, and loss solely to particle surface area (CS).

325 **4 Conclusions**

This paper reports, to the best of our knowledge, the first measurements of $H_2SO_4$ from the Amazon Basin. The median concentrations measured during both the wet (IOP 1: 7.82 x $10^5$ molec cm$^{-3}$) and dry (IOP 2: 2.59 x $10^5$ molec cm$^{-3}$)



seasons differed only slightly from each other, indicating that seasonal changes have minimal impact on $H_2SO_4$ in this region. These concentrations are consistent with measured values from the boreal forest in Hyytiälä (Dada et al., 2020; Mikkonen et al., 2011), and much lower than measurements from more urban locations (Dada et al., 2020; Mikkonen et al., 2011). Our results show minimal diurnal variation across both seasons and no clear correlation with global radiation, in contrast previous measurements of $H_2SO_4$ from a variety of locations (Dada et al., 2020; Mikkonen et al., 2011; Petäjä et al., 2009). These results suggest that photochemical oxidation of $SO_2$ by OH is not the only source of $H_2SO_4$ in the region, as well as demonstrate the importance of including measurements from a wide range of sites to develop a general-use $H_2SO_4$ proxy.

The best predictive proxy for all light conditions was the boreal proxy reported in Dada et al. (2020). This was the only radiation-dependent proxy to provide nighttime estimations, which is a clear advantage for use in an environment like this one where there is measurable nighttime $H_2SO_4$. If nighttime estimations of $H_2SO_4$ are necessary for environments similar to the Amazon Basin, the boreal proxy reported in Dada et al. (2020) is the best available estimation for low-light data when measurements of OH are unavailable. However, we note that the nighttime estimations are incomplete because the production via OH oxidation of $SO_2$ is not included. The validity of the rural proxy from Dada et al. (2020) and the best proxy from Mikkonen et al. (2011) are supported for daytime estimations (radiation > 100 W m$^{-2}$) by these results. All three provide estimations within the $25^{th}$ to $75^{th}$ percentile of the measured concentrations under these conditions.

Based on the measurements from the Amazon Basin and the proxy results, both the sCI and $SO_2$ oxidation by OH pathways for $H_2SO_4$ production contribute during low-light and nighttime conditions. This combination under low-light conditions is not currently accounted for by any existing $H_2SO_4$ proxy, and may be responsible for low-light $H_2SO_4$ in other tropical and low-$NO_x$ environments. The combination of biogenic emissions from the rain forest combined with fresh anthropogenic emissions from local farms and aged anthropogenic emissions from Manaus provides more chemical heterogeneity than what is observed in Hyytiälä (Asmi et al., 2011; Dada et al., 2017; Kulmala et al., 2016), which may help explain the observed discrepancy between the measured and estimated $H_2SO_4$ concentrations. More measurements from the Southern Hemisphere, which has lower $NO_x$ compared to the Northern Hemisphere, should be used to test and construct $H_2SO_4$ proxies to more accurately represent the variety of $H_2SO_4$ and OH sources.

These results, which are the first to test existing proxies using data from the Southern Hemisphere, demonstrate the challenges in simplifying the complex processes controlling $H_2SO_4$ levels into an equation. We observed that radiation is not always an effective substitute for OH concentrations, particularly when global radiation is between 0 - 100 W m$^{-2}$. This substitution is not valid in locations where there is measurable OH at night, due to production from secondary sources such as $O_3$ oxidation of alkenes like isoprene. While OH is difficult to measure, effort should be made to collect more measurements across a variety of environments to assess its contribution to the $H_2SO_4$ population during low-light and nighttime conditions, to help develop proxies that more accurately account for this nighttime chemistry. In particular, more OH measurements are needed in the Southern Hemisphere to constrain OH models and improve $H_2SO_4$ parameterizations.



*Data availability.* GoAmazon2014/5 data used in this study are available from the ARM website: https://doi.org/10.5439/1346559 (ARM, 2022).

*Author contributions.* Measurements were made by SK, SS, AG, RS, OVB, JT, RAFS, and JS. DM performed proxy estimations, and DM and JS helped analyze results. DM prepared the manuscript with contributions from SK, SS, AG, RS, OVB, JT, RAFS, and JS.

*Competing interests.* The authors declare that they have no conflicts of interest.

*Acknowledgements.* Institutional support for GoAmazon2014/5 was provided by the Central Office of the Large Scale Biosphere Atmosphere Experiment in Amazonia (LBA), the National Institute of Amazonian Research (INPA), and Amazonas State University (UEA) and the local Research Support Foundation (FAPEAM/GOAMAZON). We also acknowledge support from the Atmospheric Radiation Measurement (ARM) Climate Research Facility, a user facility of the United States Department of Energy, Office of Science, sponsored by the Office of Biological and Environmental Research, and support from the Atmospheric System Research (ASR, DE-SC0011122 and DE-370 SC0011115) program of that office. JS acknowledges support from a Brazilian Science Mobility Program (Programa Ciência sem Fronteiras) Special Visiting Researcher Scholarship. RS acknowledges grants RYC2020-029216-I and CEX2018-000794-S funded by MCIN/AEI/ 10.13039/501100011033 and by "ESF Investing in your future". The authors would also like to thank Michelia Dam, Hayley Glicker, Adam Thomas, and Jeremy Wakeen for their contributions to discussions regarding this project.



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
