# Peer review of "Sulfuric acid in the Amazon Basin: Measurements and evaluation of existing sulfuric acid proxies"

_Atmospheric Chemistry and Physics, 2022_

## Author Response (AR1)

We would like to thank both of the anonymous reviewers and Dr. Andreae for their thoughtful comments on our manuscript. Please read our responses below and refer to the attached manuscript, in which we show our edits to the original manuscript.

Community comment:

Upon reading this interesting paper, I would like to share some concerns:

1. The measurement site near Manacapuru is located downwind of the city of Manaus, and is thus alternatingly within the Manaus urban plume or in background air with only minor anthropogenic inputs. Trace gas and aerosol concentrations vary greatly between these conditions, as shown by Kuhn et al. (2010; not cited here) and several papers from the GoAmazon team. One would thus expect to find different concentrations of the species discussed here and it would seem essential to me to discuss these conditions separately.

   *We thank the commentator for this suggestion and have incorporated analysis of trace gas concentrations and proxy estimates to compare periods with and without influence from Manaus. This is done in starting at line 205, and in the Supporting Information section S4. We have also added a reference to the Kuhn et al. manuscript.*

2. In the Methods section, the detection limit of the $SO_2$ analyzer is given as $2.4 \times 10^8$ $cm^{-3}$. As 1 ppt corresponds to about $2 \times 10^7$ molec $cm^{-3}$ at sea level, this would correspond to about 12 ppt. In contrast, the detection limit given by the manufacturer is 0.1 ppb, and that stated in Springston (2016) is 0.3 ppb for 60 sec averages. The $SO_2$ concentrations in Table 2 show median values around $1.5 \times 10^9$, or about 75 ppt, which would be well below the stated detection limit of the instrument.

   *This was an error on our end. We have updated Table 2 and subsequent discussion to reflect the correct measurements of SO2 made during the campaign, which had previously been used in the parameterizations.*

3. In their comparisons with previous work at other sites, the authors use the term "consistent". It is not clear to me what "consistent" means in this context. Do they mean comparable, identical, similar? Would a factor two difference still be consistent? I recommend that instead of using such vague terminology, the authors provide quantitative comparisons, ideally in the form of a table.

   *We agree with the commentator that we could have overzealously used the word "consistent" in this manuscript. We analyzed each use of the word and have either left it if we feel that it adequately expresses a point that does not warrant a*

*numerical comparison (e.g., "consistent with the idea that …"), substituted the word with one that is more appropriate, or left the word and added specific data to provide quantitative comparisons when possible.*

*We also liked the idea of providing a table that summarizes our observations and compares those to others that are mentioned in the text. We agree that it is a more efficient way of comparing our data to those of prior studies. Table 3 has been added and text also has been modified to refer to the table when necessary (see Section 3). This has the advantage of eliminating the need to constantly cite the papers that provide the data, since this is also summarized in the table.*

4. Line 160 and elsewhere: Rcia et al. (2000) should be Yamasoe et al. (2000).

   *We thank the commentator for this correction. The references have been updated accordingly.*

5. Line 159ff: Note that the differences between wet and dry seasons were not "hypothesized" by previous authors, but based on measurements. This has been documented in numerous publications (Artaxo et al., 2002; to name just a few; Andreae, 2009; Martin et al., 2010; Andreae et al., 2015). The lesser interseasonal difference observed here may be related to in influence of pollution from Manaus, which is present year-round.

   *We have made the suggested change.*

Andreae, M. O., Correlation between cloud condensation nuclei concentration and aerosol optical thickness in remote and polluted regions: Atmos. Chem. Phys., 9, 543–556, 2009.

Andreae, M. O., Acevedo, O. C., Araujo, A., Artaxo, P., Barbosa, C. G. G., Barbosa, H. M. J., Brito, J., Carbone, S., Chi, X., Cintra, B. B. L., da Silva, N. F., Dias, N. L., Dias, C. Q., Ditas, F., Ditz, R., Godoi, A. F. L., Godoi, R. H. M., Heimann, M., Hoffmann, T., Kesselmeier, J., Konemann, T., Kruger, M. L., Lavric, J. V., Manzi, A. O., Lopes, A. P., Martins, D. L., Mikhailov, E. F., Moran-Zuloaga, D., Nelson, B. W., Nolscher, A. C., Nogueira, D. S., Piedade, M. T. F., Pohlker, C., Poschl, U., Quesada, C. A., Rizzo, L. V., Ro, C. U., Ruckteschler, N., Sa, L. D. A., Sa, M. D., Sales, C. B., dos Santos, R. M. N., Saturno, J., Schongart, J., Sorgel, M., de Souza, C. M., de Souza, R. A. F., Su, H., Targhetta, N., Tota, J., Trebs, I., Trumbore, S., van Eijck, A., Walter, D., Wang, Z., Weber, B., Williams, J., Winderlich, J., Wittmann, F., Wolff, S., and Yanez-Serrano, A. M., The Amazon Tall Tower Observatory (ATTO): overview of pilot measurements on ecosystem ecology, meteorology, trace gases, and aerosols: Atmos. Chem. Phys., 15, 10723-10776, doi:10.5194/acp-15-10723-2015, 2015.

Artaxo, P., Martins, J. V., Yamasoe, M. A., Procópio, A. S., Pauliquevis, T. M., Andreae, M. O., Guyon, P., Gatti, L. V., and Leal, A. M. C., Physical and chemical properties of aerosols in the wet and dry seasons in Rondonia, Amazonia: J. Geophys. Res., 107, -, doi:10.1029/2001JD000666, 2002.

Kuhn, U., Ganzeveld, L., Thielmann, A., Dindorf, T., Schebeske, G., Welling, M., Sciare, J., Roberts, G., Meixner, F. X., Kesselmeier, J., Lelieveld, J., Kolle, O., Ciccioli, P., Lloyd, J., Trentmann, J., Artaxo, P., and Andreae, M. O., Impact of Manaus City on the Amazon Green Ocean atmosphere: ozone production, precursor sensitivity and aerosol load: Atmos. Chem. Phys., 10, 9251-9282, doi:10.5194/acp-10-9251-2010, 2010.

Martin, S. T., Andreae, M. O., Artaxo, P., Baumgardner, D., Chen, Q., Goldstein, A. H., Guenther, A., Heald, C. L., Mayol-Bracero, O. L., McMurry, P. H., Pauliquevis, T., Pöschl, U., Prather, K. A., Roberts, G. C., Saleska, S. R., Dias, M. A. S., Spracklen, D., Swietlicki, E., and Trebs, I., Sources and properties of Amazonian aerosol particles: Rev. Geophys., 48, RG2002, doi:10.1029/2008RG000280, 2010.

Yamasoe, M. A., Artaxo, P., Miguel, A. H., and Allen, A. G., Chemical composition of aerosol particles from direct emissions of vegetation fires in the Amazon Basin: water-soluble species and trace elements: Atmospheric Environment, 34, 1641-1653, 2000.

Reviewer 1 comments:

I am curious why from Dada et al. (2020) that the Criegee term does not include a sink for Criegee intermediates. The fate of Criegee Intermediates would depend on RH and perhaps concentrations of organic acids.

We agree with the review that this is an area to address when including sCI production in a parameterization. We qualify our use of this proxy in this region with high RH in the updated manuscript on page 17, line 383: "*Additionally, this parameterization does not include a sink for Criegee intermediates, which may be important in this region with high RH.*"

Similar to that comment: I expect that the Criegee + SO2 rate to be quite dependent on alkene type, which may be significantly different between boreal and tropical forests. How much does changing the coefficients in Proxy 4 improve the estimations?

Changing the coefficients in Proxy 4 does not improve the estimations During both IOPs, data points from daylight (> 100 $Wm^{-2}$) conditions fall on the 1:1 line, but the low-light points do not (Fig. 5) Adjusting the coefficients in Proxy 4 did not improve the estimates from low-light conditions while leaving the daylight estimations intact. The second suggestion in regards to adjusting Proxy 4 resulted in improved estimates if we substitute OH for radiation, as we describe in response to the next comment.

Since the authors believe that secondary OH production (especially under low-light or nighttime conditions) is likely underestimated, why can't the OH concentrations measured directly be used in Proxy 4 to see if how much that improves estimation?

We thank the reviewer for this suggestion and have tested Proxy 4 with OH concentrations. Not surprisingly, this proxy provided much improved estimates of sulfuric acid during low-light hours, though there are still some low-light data points that also corresponded to low levels of OH. This discussion has been added to the text (see lines 318ff and 323ff).

Related to my previous comment: if OH itself cannot explain the discrepancy, I wonder if there are other SO2 oxidation pathways that need to be taken into account. There is a recent boom in SO2 oxidation literature and proposed mechanisms. Some of these mechanisms (will need to be homogeneous) may be applicable. This is speculative and depends on what the authors may find from my previous suggestion.

Based on our results from the previous point, we hypothesize that the underestimations seen at night might be due to other sulfur sources, such as dimethylsulfide, hydrogen sulfide, and mehtylmercaptan, all of which have been previously measured in the Amazon Basin. This was added into the text (see lines 283ff).

Minor comments:

In Figure 2, the R2 for both Proxy 1 and Proxy 2 are 0.46. Is this a coincidence or is there a mistake? It may be useful to show the slopes of the regression too.

We thank the reviewer for this comment – it was a typo, which has been updated in the manuscript. The slopes of the fits have also been added to the corresponding discussion.

I prefer the × symbol over the letter x in scientific notation.

This change has been made to the manuscript.

Reviewer 2 comments:

General comments

As suggested by Referee #1, I also would recommend adding tests of the predictions from Proxy 4 using the measured OH instead of Global radiation (with coefficients of Proxy 4 fitted according to the data used in this study). This way the authors could discuss how much including the measured night-time OH oxidation of SO2 would improve the proxy concentrations.

We thank the reviewer for this suggestion and have tested Proxy 4 with OH concentrations. Not surprisingly, this proxy provided much improved estimates of sulfuric acid during low-light hours, though there are still some low-light data points that also corresponded to low levels of OH. This discussion has been added to the text (see lines 318ff and 323ff).

Minor comments and technical corrections

**Lines 105-107: What is the reason of the slight changes in the particle size range measured by the SMPS?**

We don't know the reason for this change, but suspect that it came as a result of a recalibration of flows or voltages of the SMPS. Regardless, the dataset that we used and referenced in this manuscript has been QCed but the DOE ARM instrument mentors and we are confident in its accuracy.

Table 1: Typo in Proxy 3, the exponent of [SO2] should be 0.62 (not -0.62)

The typo has been fixed.

Line 142: Median of H2SO4 concentration during IOP 2 is given in the text as $2.56 \cdot 10^5$ cm$^{-3}$, whereas in Table 2 it is $2.59 \cdot 10^5$ cm$^{-3}$. Please check which one is correct.

This typo has been fixed and updated in Table 2 as well as the text.

Lines 160-162: This is unclear sentence, what are the median values reported at the end of this sentence?

The text has been updated to clarify that the reported values are median SO2 concentrations.

Line 186: Explain more clearly, what do you mean with "36% of the total H2SO4 was measured at night"? Do you mean that the level of night-time concentrations was 36% of the daytime concentrations or what?

This line was changed to clarify that the total nighttime concentrations were 36% of the total measured concentrations of H2SO4.

Line 245: In the particle number size-distributions measurements by SMPS, is the sample air dried, and therefore taking the RH into account in the proxies would be more representative of the actual sink term? This information could be added to Section 2.2.2 where the SMPS measurements are described.

The sample air is dried to a maximum of 20 % RH, which has been added to Section 2.2.2 per the reviewer's suggestion. We have added this information into our analysis of the Proxy 3 results as well.